# Transcriptome Analysis Reveals the Molecular Mechanisms of Carrot Adaptation to Alternaria Leaf Blight

**DOI:** 10.3390/ijms252313106

**Published:** 2024-12-06

**Authors:** Chen Liang, Donghang Zhao, Chenggang Ou, Zhiwei Zhao, Feiyun Zhuang, Xing Liu

**Affiliations:** State Key Laboratory of Vegetable Biobreeding, Institute of Vegetables and Flowers, Chinese Academy of Agricultural Sciences, Beijing 100081, China; lc010206@163.com (C.L.); 15027438324@163.com (D.Z.); ouchenggang@caas.cn (C.O.); zhaozhiwei@caas.cn (Z.Z.)

**Keywords:** *Daucus carota*, adaptation mechanism, resistance gene, transcriptome

## Abstract

Carrot (*Daucus carota* L.) is an important vegetable crop that is rich in carotenoids and is widely cultivated throughout the world. Alternaria leaf blight (ALB), caused by infection with *Alternaria dauci* (*A. dauci*), is the most serious fungal disease in carrot production. Although several quantitative trait loci associated with ALB resistance have been identified, the genetic mechanisms underlying this resistance remain largely unelucidated. The aim of the present study was to clarify the infection mode of *A. dauci* and examine the molecular mechanisms underlying carrot cultivar adaptation to ALB by RNA sequencing. Microscopic observation revealed that *A. dauci* invades leaf tissues by entering through stomata, and resistant germplasms may significantly inhibit the infection and colonization of *A. dauci*. In addition, transcriptomic analyses were performed to detect the key pathways and genes associated with the differential responses between ALB-resistant (HB55) and ALB-susceptible (14088) carrot cultivars. These results suggest that the secondary metabolic process, phenylpropanoid biosynthesis, and tyrosine metabolism might play important roles in the resistance of carrots to *A. dauci*. Three candidate genes (*LOC108208301*, *LOC108215577*, and *LOC108224339*) that were specifically upregulated in the resistant carrot cultivar ‘HB55’ after *A. dauci* infection were identified as the key resistance response genes. These findings provide insights into the resistance mechanism of carrots to ALB, as well as key candidate genes and information on expression regulation for the molecular breeding of carrot disease resistance.

## 1. Introduction

Carrot (*Daucus carota* L.) is a biennial dicotyledonous plant in the *Umbelliferae* family, and is widely grown in Asia, Europe, and the Americas [1]. The taproot of carrot is the greatest source of beta-carotene in the human diet [2,3]. China is the world’s top carrot producer, with its cultivation area covering over 400,000 hectares, which is approximately one-third of the global total (FAO Statistics; see URLs). The provinces of Henan, Hebei, Shanxi, Anhui, Shandong, Fujian, and Inner Mongolia are the main cultivation areas in China, and commercial hybrid carrot varieties are widely planted in these areas, with an area close to 35,000 hectares.

Alternaria leaf blight (ALB) is a major global disease of carrots. The disease is present worldwide in all carrot growing areas and is the most common destructive foliar disease on this crop [4]. It was first discovered in Germany in 1855, and has since spread to areas of carrot production around the world [5]. Inoculum sources include mycelium in over-wintering carrot debris, diseased volunteer or wild carrots, and infected seeds [6]. However, no immune germplasm against ALB has been discovered to date. At least eight diseases that impact carrots have been reported in China, among which ALB has become a major epidemic disease in large-scale planting areas, especially in the northern provinces of China. ALB not only affects the photosynthetic activity of carrot leaves and carbohydrate synthesis but also reduces the efficiency of mechanical harvesting by causing leaf necrosis, resulting in a 40–60% reduction in carrot yield in severe cases [7,8]. In addition, many chemical pesticides are applied to control this disease, causing environmental pollution and endangering the safety of consumers.

*Alternaria dauci* (*A. dauci*)*,* a large-spore *Alternaria* species, is the causal agent of ALB [9]. In the early stage of the disease, *A. dauci* infects the above-ground leaves and petioles of carrots, forming semicircular or irregularly shaped brown or black lesions on the tips or margins of the leaves. The lesions then extend to the entire leaf, causing the leaf blade to lose its green color. In severe cases, the entire leaf blade and above-ground parts wilt and die. The damaged carrot leaf tissue loses its ability to photosynthesize, leading to a decrease in water absorption and nutrient accumulation in the taproots. Carrots that are grown in warm, humid environments are highly susceptible to ALB. In warm environments, pathogens are metabolically active and grow faster, allowing them to produce spores more efficiently. The spores of *A. dauci* are transmitted via a variety of routes, including seeds, plant debris, soil, air, and water, making the control of ALB difficult [10,11]. By using electron microscopy, Lubaina and Murugan discovered that when *Alternaria* penetrates directly through the epidermis or stomata, hyphal branches grow in the intercellular space of mesophyll tissue and cause cell and cell wall damage [12]. However, the ultrastructural changes in carrot leaf tissue caused by *A. dauci* have not been studied. Chemical fungicides play only a limited role in preventing and controlling ALB outbreaks and the early stages of the disease, and the most fundamental way to prevent and control the ALB is by identifying disease resistance genes and breeding disease-resistant varieties [13]. Le Clerc et al. identified three QTLs associated with resistance to ALB in F_2_ progenies obtained from a cross between the susceptible line S269 and the partially resistant line R268 firstly [14], and further detected eleven QTLs associated with ALB resistance using SSR markers [15]. However, to effectively guide breeding programs, the mechanisms underlying these QTLs still need to be deciphered. In addition, Koutouan et al. used a combination of targeted and non-targeted metabolomics to compare the leaf metabolome of four carrot genotypes with different resistance levels. The results indicated that certain metabolites, including flavones, and mono- and sesquiterpenes, may play a role in determining resistance or susceptibility [16].

In nature, plants are constantly exposed to pathogens. With the rapid development of molecular genetics, molecular biology, and bioinformatics, analyzing complex host–pathogen interaction networks has become possible [17,18]. RNA interference techniques, especially host-induced gene silencing (HIGS) and spray-induced gene silencing (SIGS), have shown potential in plant pathogen defense and control [19,20,21]. Gene editing technology, especially CRISPR/Cas9, has greatly promoted functional research into resistance genes [22,23], and these technologies are now being used to improve disease resistance in plants [24]. In addition, the application of high-throughput technologies such as large-scale transcriptomics, proteomics, and metabolomics has provided rich data for studying plant immunity [25,26]. In particular, many differentially expressed genes (DEGs) and enrichment pathways have been identified through transcriptome analysis. These DEGs are usually related to the immune response, and provide clues to the key adaptability mechanisms of disease resistance [27,28]. In addition, some DEGs may play critical regulatory roles and become key targets for improving disease resistance [29,30].

In a recent study, Sun et al. analyzed transcriptome data obtained from FORL-resistant (19,912) and FORL-susceptible (Moneymaker) tomato cultivars [31]. There were 189 DEGs in the “plant–pathogen interaction” network, and these DEGs were subjected to Gene Ontology (GO) and Kyoto Encyclopedia of Genes and Genomes (KEGG) enrichment analyses. The results revealed that major metabolic pathways associated with plant disease resistance, including phenolic substances and endogenous hormone metabolism, carbon metabolism, the autophagy pathway, and cell wall degradation, were enriched in the MAPK and WRKY genes. Gogoi et al. analyzed the transcriptome profiles of strawberries with two resistant genotypes and a susceptible genotype after inoculation with *Phytophthora cactorum* [32]. Differential gene expression analysis revealed several defense-related genes that are highly expressed in the strawberries with the resistant genotypes relative to those with the susceptible genotype in response to *P. cactorum* infection, such as typical disease resistance genes encoding receptor-like proteins, receptor-like kinases, nucleotide-binding sites, and leucine-rich repeat proteins. Chen et al. compared the transcriptomes of resistant and susceptible pakchoi cultivars in response to *Hyaloperonospora brassicae* infection [33]. GO and KEGG pathway analyses revealed that 1073 disease resistance-related DEGs were involved mainly in plant–pathogen interactions, plant hormone signal transduction, and other photosynthesis-related metabolic processes. All of the above cases illustrate the important role of transcriptome analysis in elucidating the regulatory mechanisms of disease resistance in different plants.

To improve the resistance of carrot cultivars to ALB through molecular marker-assisted selection or genetic engineering, it is necessary to identify candidate resistance genes and clarify the resistance network as much as possible. However, the gene expression pattern in carrots following *A. dauci* infection has not been reported thus far. The mechanisms of resistance to ALB in carrots urgently need to be investigated. In a previous study, we established an artificial inoculation identification system for ALB and identified resistant germplasms in carrots [34]. The purpose of this study was to clarify the infection mode of *A. dauci* through microscopic observation and to investigate the gene expression characteristics of carrots in response to ALB by RNA sequencing. The results will lay the foundation for revealing the interaction mechanism between carrots and *A. dauci*, and provide genetic information for carrot disease resistance breeding.

## 2. Results

### 2.1. Phenotypic Analysis of the Response of Carrots to A. dauci Inoculation

To assess the resistance of ‘HB55’ and ‘14088’ to *A. dauci*, disease identification was carried out during the seedling and adult stages through manual inoculation and field nursery testing, respectively. During the seedling inoculation trial, leaves of the two varieties were observed at 0, 1, 3, 5, and 7 days post inoculation (dpi). The lesion count and lesion area results revealed that ‘14088’ exhibits rapid disease progression, while ‘HB55’ effectively resists the development of ALB (Appendix A). Three days after inoculation, black lesions appeared on the leaves of ‘14088’ seedlings. As the disease progressed, these lesions expanded, the petiole gradually turned yellow at 5 dpi, and the marginal leaves appeared dry at 7 dpi. This progression culminated in pronounced leaf yellowing and even wilting. In contrast, ‘HB55’ seedlings presented only a few lesions on their leaves until 7 dpi, and the leaves and petioles remained green (Figure 1B).

In the adult stage, no significant phenotypic alterations were observed in ‘HB55’ in the natural disease nursery. However, adult ‘14088’ plants displayed substantial leaf yellowing and wilting (Figure 1A). The disease index (DI) was calculated at the seedling and adult stages. The DI of ‘HB55’ was consistently and significantly lower than that of ‘14088’ across both growth stages (Figure 1C,D). In addition, according to the disease resistance level criteria, the average DI values of ‘HB55’ at the seedling stage and adult stage were 28.84 and 29.33, respectively, which reached the R level. The average DI values of ‘14088’ at the seedling stage and adult stage were 91.35 and 97.31, respectively, which reached the HS level (Appendix A). Collectively, these observations reveal substantial disparities in ALB resistance between ‘HB55’ and ‘14088’, making them suitable materials for investigating the resistance mechanism of carrots to ALB.

### 2.2. Scanning Electron Microscopy Analysis of Leaves from the Two Carrot Cultivars

The invasion and colonization of *A. dauci* were characterized via SEM. Leaves of both ‘HB55’ and ‘14088’ were observed at different times after inoculation. In ‘14088’, mycelia were discernible on the surface of the leaves at the earliest stage of infection (1 dpi), and their proliferation was evident at 3 dpi. Moreover, the mycelia had formed distinct colonies that covered almost the entire leaf at 5 dpi. Finally, at 7 dpi, severe pore damage had occurred on the leaf surface, causing severe disease symptoms, and mature conidia were observed in the pores (Figure 2A). Furthermore, ‘HB55’ presented strong resistance to *A. dauci* invasion at the early stage of inoculation (Figure 2B). Mycelia were not observed on the leaves of ‘HB55’ at up to 3 dpi, indicating that the resistant cultivar was able to delay the invasion of the pathogen. At 7 dpi, although mycelia were present on the surface of the leaves, there was no pore destruction, indicating that ‘HB55’ could effectively inhibit further colonization of *A. dauci* on the leaves and prevent leaf damage. The above results were also corroborated by the statistical analysis of mycelial infestation areas in both materials at different times after inoculation (Appendix A).

Mycelia were not observed around aqueous pores on the leaf surface after inoculation in either the susceptible or the resistant cultivar. Interestingly, mycelia were observed around the stomata on the leaves of the susceptible cultivar after inoculation. Fungal spores with an “attached spore structure” entered the stomata at 3 dpi and produced conidia at these locations (Figure 2A). Therefore, the invasion of *A. dauci* relies mainly on stomata.

### 2.3. Transcriptome Analysis

To analyze the transcriptional changes in carrots in response to *A. dauci* infection, RNA-seq was conducted on different carrot cultivars. Leaves of the ‘14088’ and ‘HB55’ cultivars were collected at 0, 3, and 7 dpi, with three biological replicates at each time point. The samples of the resistant carrot variety ‘HB55’ are represented by R0, R3, and R7, whereas the samples of the susceptible carrot variety ‘14088’ are represented by S0, S3, and S7. The cDNA libraries were constructed and sequenced using the Illumina NovaSeq 6000 platform. A total of 801,671,762 raw reads were generated from 18 samples, and 791,730,930 high-quality clean reads were identified after filtering. In total, 119.4 Gb of clean data were obtained, with an average of 43,985,052 clean reads per sample. The percentage of Q20 bases ranged from 98.83% to 99.04%, the percentage of Q30 bases ranged from 96.7% to 97.2%, and the GC content was approximately 43.87% (Appendix A). These findings indicated that the transcriptome data were of a high quality and could be used for further data analysis.

Filtered clean reads were compared to the latest carrot reference genome (DH1 v 3.0) using HISAT2 software (http://daehwankimlab.github.io/hisat2/ (accessed on 8 April 2024), v2.1.0), which revealed 88.30% alignment [35]. Reads mapped to the reference genome accounted for 95.78% of these reads (Appendix A). The principal component analysis (PCA) results revealed high reproducibility among samples within the group. The different groups were separated from each other (Figure 3A), with PCA 1 and PCA 2 clearly segregating different samples and differentiating between the cultivars. Correlation analyses of gene expression levels between samples showed a relatively high degree of reproducibility among samples within the groups (Appendix A), ensuring the reliability of the subsequent differential gene analyses.

#### 2.3.1. Identification of DEGs

In this study, a total of 13,969 DEGs were identified. Taking S0 and R0 as the CK groups, 9037 DEGs were found in ‘14088’ and 14,745 DEGs were found in ‘HB55’. In ‘14088’, 1443 and 3471 DEGs were upregulated, and 876 and 3247 DEGs were downregulated at 3 dpi and 7 dpi, respectively. In ‘HB55’, 3061 and 4004 DEGs were upregulated, and 3211 and 4469 DEGs were downregulated at 3 dpi and 7 dpi, respectively. In both ‘14088’ and ‘HB55’, the number of DEGs at 7 dpi was greater than that at 3 dpi. Interestingly, there were more upregulated DEGs than downregulated DEGs at 3 dpi and 7 dpi in ‘14088’ and more downregulated DEGs than upregulated DEGs at 3 dpi and 7 dpi in ‘HB55’. The DEGs between the S3 and S7 samples were identified, of which 913 were downregulated and 1507 were upregulated. A total of 1481 DEGs were identified between the R3 and R7 samples, of which 571 DEGs were downregulated and 910 were upregulated (Figure 3B). In addition, a total of 2948, 3218, and 2645 DEGs were identified between the S0 and R0 samples, the S3 and R3 samples, and the S7 and R7 samples, respectively (Appendix A).

As shown in the Venn diagram, 691, 2330, and 219 DEGs were specifically expressed in the R0_vs_R3, R0_vs_R7, and R3_vs_R7 comparison groups, respectively. In addition, 524 DEGs were expressed in all of the groups (Figure 3C). Additionally, 698 DEGs were expressed in the S0_vs_S3, S0_vs_S7, and S3_vs_S7 comparison groups (Figure 3D).

#### 2.3.2. Functional Enrichment Analysis of DEGs Identified in Carrot Cultivars Infected with *A. dauci*

The functions of the 524 DEGs expressed in all three comparison groups in ‘HB55’ were investigated. We annotated the DEGs using the Gene Ontology (GO) database and performed enrichment analysis. We identified the top 10 GO terms with the smallest *p*-values that were significantly enriched in the categories of biological process (BP), cellular component (CC), and molecular function (MF) (Figure 4A). The GO term with the smallest *p*-value in the BP classification was the unsaturated fatty acid biosynthetic process (GO:0006636), with a DEG count of 13. The secondary metabolite biosynthetic process (GO:0044550) was also enriched in 17 DEGs. The GO term with the smallest *p*-value in the CC classification was the apoplast (GO:0048046), with a DEG count of 20. The term with the smallest *p*-value in MF was oxidoreductase activity, acting on paired donors (GO:0016717), with 13 DEGs (Appendix A).

To further clarify the roles of the DEGs, enrichment analysis of 524 DEGs was performed using the Kyoto Encyclopedia of Genes and Genomes (KEGG) database. We performed statistical and enrichment analyses on the top 10 KEGG pathways with the smallest *p*-values in all the comparison groups. Phenylpropanoid biosynthesis (ko00940) was enriched in 18 DEGs. In addition, tyrosine metabolism (ko00350) and flavonoid biosynthesis (ko00941) were enriched in four and five DEGs, respectively (Appendix A). The KEGG results suggested that phenylpropanoid biosynthesis (ko00940), phenylalanine, tyrosine and tryptophan biosynthesis (ko00400), tyrosine metabolism (ko00350), and flavonoid biosynthesis (ko00941), which were previously reported to be associated with ALB resistance, were significantly enriched (Figure 4B).

The GO analysis of the 698 DEGs in the three comparison groups in ‘14088’ revealed 10 BP terms, among which the unsaturated fatty acid biosynthetic process (GO:0006636), response to chitin (GO:0010200), and the lignin metabolic process (GO:0009808) were the most enriched (Figure 5A). There were 10 terms related to cell composition, among which DEGs were most significantly enriched in the plastid thylakoid membrane (GO:0055035), chloroplast thylakoid membrane (GO:0009535), and obsolete extracellular region part (GO:0044421). The term with the smallest *p*-value in the MF category was oxidoreductase activity, acting on paired donors (GO:0016717) (Appendix A).

Next, 698 DEGs were mapped to Kyoto Encyclopedia of Genes and Genomes (KEGG) pathways (Figure 5B). Among them, phenylpropanoid biosynthesis (ko00940), MAPK signaling pathway–plant (ko04016), and plant–pathogen interaction (ko04626) were enriched in the DEGs (Appendix A).

#### 2.3.3. Candidate Genes Involved in ALB Resistance in Carrots

To further clarify the relevant genes involved in the disease resistance response, correlation heatmaps were constructed for the DEGs involved in plant disease resistance. Fourteen DEGs involved in the secondary metabolic process pathway were continuously highly expressed after inoculation with *A. dauci* in ‘HB55’, whereas the expression of three DEGs continuously decreased. Among the fouteen DEGs whose expression increased, two specific DEGs, *LOC108208301* and *LOC108215577*, were upregulated in ‘HB55’, whereas these genes were unchanged in ‘14088’ (Figure 6A). Similarly, *LOC108215577,* which is involved in the phenylpropanoid biosynthesis pathway, was also specifically upregulated in ‘HB55’ but remained unchanged in ‘14088’ (Figure 6B). In addition, we found that in ‘HB55’, *LOC108224339* was enriched in both the phenylalanine, tyrosine, and tryptophan biosynthesis pathway and the tyrosine metabolism pathway. Interestingly, in contrast to the unchanged expression of *LOC108224339* in ‘14088’, this gene was highly expressed in ‘HB55’ (Figure 6C,D).

### 2.4. qRT-PCR Validation of the Transcriptomic Data

To validate the RNA sequencing data, gene expression analyses were conducted in the same samples via qRT–PCR (Figure 7). Nine DEGs were selected, and the primers for these genes were developed via Primer 5.0 software (Appendix A). After inoculation, the expression levels of *LOC108227809*, *LOC108199362*, *LOC108210565*, *LOC108209160*, *LOC108206716*, *LOC108210412,* and *LOC108194157* in ‘HB55’ and ‘14088’ were all increased compared to those at 0 dpi. The expression level of *LOC108211710* decreased in ‘HB55’ after inoculation compared to that at 0 dpi, while the expression level in ‘14088’ was unchanged compared to that at 0 dpi. *LOC108224339*, a candidate gene involved in ALB resistance, was significantly upregulated in ‘HB55’ after inoculation, whereas it remained unchanged in ‘14088’. The high consistency between the qRT-PCR results and transcriptomic data indicated the accuracy of the transcriptome sequencing results.

## 3. Discussion

ALB is one of the most common and destructive diseases that affects carrots worldwide [36]. With the increase in severe weather conditions such as continuous rainfall and high temperatures in recent years, the damage caused by ALB has also shown a trend of expansion and increased severity in recent years. Consistent with the treatment of other fungal diseases in different crops, breeding disease-resistant varieties is the most fundamental way to control ALB. These varieties can sustain high levels of resistance even under conditions of high temperatures and humidity [13]. Notably, the cultivation of disease-resistant varieties can also reduce the damage to the environment caused by the application of a large number of chemical fungicides [37]. However, disease-resistant germplasm resources must be identified to breed plants with increased disease resistance [38,39]. On this basis, the use of the resistant materials for multiomics analysis has been proven to be an effective way to elucidate disease resistance mechanisms [40,41]. Initially, ALB-resistant resources were not available, and the resistances of existing domestic and foreign carrot varieties are generally insufficient to meet the needs of the industry [42]. Recently, a carrot resistant line, ‘HB55’, was identified through an artificial inoculation identification system [34]. Thus, to identify target genes and pathways for future functional research and molecular breeding efforts, comparative analysis of resistant and susceptible lines can be conducted, including factors such as disease symptoms and transcriptional expression.

Plants are frequently exposed to various stresses that affect their growth and development [43]. Approximately 85% of plant diseases are caused by fungi, which constitute the largest number of plant pathogens. Plant stomata are the main point of entry for many pathogens [44,45]. In the present study, we used SEM to examine leaves of ‘HB55’ and ‘14088’ at 0, 1, 3, 5, and 7 days after inoculation with *A. dauci*. Observations of the leaves of ‘14088’ revealed that as the disease progressed, many of mycelia and attached spores appeared around the stomata of the leaves in contrast to the aqueous pores, where mycelia and spores were not present. It was concluded that *A. dauci* infects carrot plants mainly through stomata on the abaxial surface of the leaves. A comparison of the leaves of resistant and susceptible cultivars after inoculation revealed that the susceptible cultivar was less resistant to *A. dauci,* with mycelia colonizing the leaves at 3 dpi. At 7 dpi, obvious symptoms of ALB were detected. In contrast, disease-resistant material was strongly resistant to pathogen colonization, maintaining leaves’ integrity and effectively inhibiting disease development. Here, the invasion of *A. dauci* and the key time points for disease development after inoculation have been clarified for the first time, providing a reference for subsequent research on carrot resistance to ALB.

Elucidating the mechanisms underlying the resistance of carrots to *A. dauci* infection is critical for refining integrated management strategies against ALB. This study assessed the differential gene expression between the resistant cultivar ‘HB55’ and the susceptible cultivar ‘14088’ after *A. dauci* infection. In ‘HB55’, 524 DEGs were identified across the three comparison groups (R0_vs_R3, R0_vs_R7, and R3_vs_R7) and further characterized. Similarly, 698 DEGs were identified in the three ‘14088’ comparison groups (S0_vs_S3, S0_vs_S7, and S3_vs_S7) and further characterized. Phenylpropanoids are not only indicators of plant stress responses to various light or mineral treatments but also key mediators of plant resistance to pests [46,47]. Tyrosine is an aromatic amino acid that is the precursor of various plant metabolites with different physiological functions [48,49]. KEGG pathway enrichment analysis of DEGs revealed two common pathways in the comparison of resistant and susceptible cultivars, of which “phenylpropanoid biosynthesis” and “tyrosine metabolism” were significantly enriched. Analysis of DEGs enriched in these pathways revealed that some genes were upregulated in ‘HB55’ but unchanged in ‘14088’. These findings suggest that “phenylpropanoid biosynthesis” and “tyrosine metabolism” may be closely related to the resistance of carrots to *A. dauci* infection. The resistance of plants to pathogenic microorganisms is largely dependent on the synthesis of disease-resistant secondary metabolites [50]. A multitude of secondary metabolites, together with the enzymes responsible for their biosynthesis and degradation, are directly involved in plant resistance to biotic stresses [51,52]. GO enrichment analysis revealed that 17 DEGs in ‘HB55’ were involved in secondary metabolite biosynthetic processes.

Plant immune research has shown that mitogen-activated protein kinase (MAPK) signal transduction, transcription factors such as NAC, MYB, and WRKY, pathogenesis-related (PR) genes, phytohormone metabolism, and signal transduction play important roles in the plant defense system independently or cooperatively [53,54]. In particular, the roles of phytohormones, salicylic acid (SA), jasmonic acid (JA), and ethylene (ET) in host–pathogen interactions have been widely reported [55,56], and the functions of abscisic acid (ABA), auxins, gibberellic acid (GA), cytokinins (CK), and brassinosteroids (BRs) in plant defense have also been elucidated [57,58,59,60,61]. Therefore, the role of plant hormones in the resistance of carrots to ALB should not be ignored. The inclusion of earlier sampling time points or the supplementation of other materials with different genetic backgrounds should be considered in subsequent studies. In addition, the cell walls of plants consist of highly complex cellulose, hemicellulose, and pectinoglycan structures, which are important defenses that pathogenic bacteria need to break through to successfully invade the plants [62]. This study clarified the differences in *A. dauci* colonization of leaves between resistant and susceptible lines through microscopic observation. However, transcriptome analysis did not reveal DEGs involved in cell wall synthesis and degradation processes. Moreover, to determine the effects of cell wall tissue characteristics, the differences in structure between resistant and susceptible lines, as well as their changes after pathogen invasion, should also be observed and monitored more precisely.

Cytochrome P450 genes are widely involved in the plant disease resistance response. For example, when the chili pepper *CaCYP1* gene was knocked out, the mutant plant lost its basal defense response against to *Xanthomonas axonopodis* [63]. In this work, three candidate genes were identified by identifying the DEGs that were activated specifically in ‘HB55’ with continuous high expression. One of the candidate genes, *LOC108208301*, also encodes a cytochrome P450 protein. Two other genes, *LOC108215577* and *LOC108224339*, encode caffeic acid 3-O-methyltransferase protein and nicotianamine aminotransferase 1 protein, respectively. The recombinant caffeic acid 3-O-methyltransferase gene was cloned and characterized from Neem. This gene is involved in ferulic acid biosynthesis, a key intermediate component of lignin biosynthesis [64,65]. Nicotianamine (NA) plays a crucial role in the transport of metal ions, including iron (Fe), in plants. Iron is an essential element required for plant growth and development [66,67]. Therefore, these three genes may play key roles in the defense response of ‘HB55’. In the future, by targeting these candidate genes and key regulatory genes in related pathways, gene function analysis and molecular marker development will accelerate the process of improving the molecular resistance of carrot varieties’ resistance to ALB.

## 4. Materials and Methods

### 4.1. Fungal Strain Culture

A single conidial isolate of *A. dauci* (acquired and previously maintained by our group) was cultured on potato dextrose agar (PDA) at 28 °C in the dark for 7 days to obtain vigorously growing colonies [68,69]. Mycelial plugs (6 mm in diameter) were cut from the actively growing leading edge of a single spore colony of *A. dauci* via a sterile hole punch, transferred to V8 planar media (200 mL of V8 juice, 3 g of CaCO_3_, 20 g of agar, and 800 mL of SDW) [70], and then cultured in a light incubator (D-82152, Friocell, Germany) at 28 °C with a light/dark cycle of 8/16 h for 10 days to obtain a large number of spores. The concentration of the spore suspension was adjusted to 1 × 10^4^/mL, and stored in a refrigerator (BCD-458WDVMU1, Haier, Qingdao, China) used for subsequent inoculation.

### 4.2. Plant Materials and Artificial Inoculation

This study was conducted on two carrot cultivars with significant differences in the disease index, namely, the resistant cultivar ‘HB55’ and the highly susceptible cultivar ‘14088’. Carrot seeds from the two different varieties were selected and sterilized via a 75% alcohol and 20% sodium hypochlorite solution. The sterilized seeds were sown in 50 hole trays with the nutrient substrate (organic matter ≥ 20%) [71]. The experiment was conducted in the light incubator at the Institute of Vegetable and Flower, Chinese Academy of Agricultural Sciences (Beijing, China), with day and night temperatures of 25 °C and 20 °C, respectively, with a cycle of 12 h/day and 12 h/night and a relative humidity of 70–80%, with a light intensity of 200~400 μmol m^−2^ s^−1^ photosynthetic photon flux density (PPFD).

To identify the resistance of ‘HB55’ and ‘14088’ to *A. dauci*, both artificial inoculation and field nursery testing were conducted during seedling and adult stages, respectively. When the two lines of carrot seedlings had 8 true leaves, the artificial inoculation with the CALB1 isolate was carried out as previously described using the leaf surface spray method [34]. The phenotypic responses were documented at intervals of 0, 1, 3, 5, and 7 days post inoculation (dpi), the size and number of lesions on the leaf surfaces were determined. Disease index (DI) was calculated at 10 dpi based on the scale as previously studied [42]. For field nursery testing, the carrot lines were planted in the nursery, located in Ulanqab City, Inner Mongolia Autonomous Region, China, and the disease severity was measured at 90 days after sowing according to the same scales. For both artificial inoculation and field nursery testing, three independent biological replicates were set, and each replicate consisted of 20 susceptible plants and 20 resistant plants.

### 4.3. Microscopic Observation of Leaves at Different Times After Inoculation

At 0, 1, 3, 5, and 7 dpi, a total of 90 leaf samples were selected from ‘HB55’ and ‘14088’ (9 leaf samples from 3 individual plants for each line at each time point). The areas of mycelial infection were counted on the leaf surfaces of the leaf samples. Meanwhile, for microscopic observation, the leaf samples were cut into 0.3 cm × 0.5 cm sizes with a scalpel blade, and quickly put into 2.5% glutaraldehyde fixative (0.1 M, pH 7.0) and stored at 4 °C for 24 h. The fixed samples were dehydrated in a gradient ethanol series (30, 50, 70, 90, and 95%) for 15 min each, and then in 100% ethanol for 15 min three times. Samples were dried in a K850 critical point dryer (Quorum Technologies, East Sussex, England) under CO_2_, mounted on copper stubs with double-sided sticky tape, and coated with goldpalladium (200 nm thickness) in a Q150V Plus sputter coater (Quorum Technologies, East Sussex, UK) [72]. The surface stomata, aqueous pores, and surface morphology of these samples were observed and photographed under a SU8010 scanning electron microscope (Hitachi High-Technologies Corporation, Minato-ku Tokyo, Japan). To eliminate accidental and erroneous shots, at least 3 strict images were obtained from each slide.

### 4.4. RNA Extraction and Transcriptome Sequencing

Leaf samples from ‘HB55’ and ‘14088’ were collected at 0, 3, and 7 dpi, resulting in a total of 18 samples. Each time point contained three biological replicates, and each replicate consists of a pool of leaves from 3 plants. Total RNA was isolated from the leaf samples using the RNA prep Pure Plant Kit (Tiangen Biotech Co., Ltd., Beijing, China) according to the manufacturer’s instructions, for qRT-PCR and RNA sequencing. The RNA samples were then sent to Personal Biotechnology Co., Ltd. (Shanghai, China) for RNA sequencing, which was conducted on an Illumina NovaSeq 6000 platform (Illumina, San Diego, CA, USA). cDNA was synthesized using the fragmented mRNA as a template. Finally, 18 transcriptome libraries were constructed.

### 4.5. Mapping of Reads to the Reference Genome

Clean reads were obtained by removing the reads containing adapters and low-quality sequences. An index of the carrot reference genome was constructed using HISAT2 (v2.1.0), and paired-end clean reads were compared with the reference genome using the same software. The read count value was statistically compared to each gene using HTSeq (v0.9.1) to determine the raw expression of the gene, and the expression was normalized using FPKM (Fragments Per Kilo bases per Million fragments).

### 4.6. DEG Identification and Data Analysis

Differential gene expression analysis was performed using DESeq (v1.38.3), and the conditions for screening DEGs were as follows: multiplicity of expression differences |log2FoldChange| > 1 and significance *p*-value < 0.05. GO term and KEGG pathway enrichment analyses were carried out via omics tools (https://www.omicshare.com/tools/ (accessed on 15 April 2024)), and the significance threshold was <0.05 for the *p*-value.

### 4.7. Screening of Genes Associated with the Response of A. dauci Infection in Carrot

To identify the candidate genes involved in ALB resistance in carrots, the expression of DEGs which were enriched in the disease-resistant pathways including the secondary metabolic process pathway, the phenylpropanoid biosynthesis pathway, the phenylalanine, tyrosine and tryptophan biosynthesis pathways, and the tyrosine metabolism pathway were shown at all time points within and between cultivars. TBtools-II v1.113 software was used to construct a heatmap [73]. Candidate genes were selected that continuously increased in expression in the resistant cultivar after inoculation, while decreasing or remaining unchanged in expression in the susceptible cultivar.

### 4.8. Quantitative RT-PCR Analysis

To validate the reliability of the transcriptome data, nine genes with high differential multiplicity were randomly selected from the DEG data for qRT-PCR analysis, with GAPDH as a reference gene. The primers for these analyses were developed using Primer 5.0 software, as shown in Appendix A. The amplification reactions were performed on an ABI 7500 FAST instrument (Applied Biosystems, Carlsbad, CA, USA) and AceQ qPCR SYBR Green Master Mix (Vazyme Biotech Co., Ltd., Nanjing, China) was used following the manufacturer’s instructions. The relative expression level was analyzed using the 2^−∆∆CT^ method. Three biological and three technical replicates were used for the qRT-PCR assays.

## 5. Conclusions

In this work, ‘HB55’ and ‘14088’, two carrot germplasms with significant adaptability differences to *A. dauci*, were selected for comparative analysis, including phenotype and gene expression differences. The results indicate that the stomata are the main infection pathway of *A. dauci*; however, the resistant germplasm can inhibit their infection and colonization. The RNA sequencing data showed that the secondary metabolic process, phenylpropanoid biosynthesis, and tyrosine metabolism pathways were activated to participate in resistance regulation. Finally, three genes (*LOC108208301*, *LOC108215577*, and *LOC108224339*) were identified as candidate genes that can be targeted for improving ALB resistance by developing resistance molecular markers or conducting genetic transformation. Overall, this study provides valuable insights into the molecular mechanism of carrot resistance to ALB, identifying key genes and pathways for potential use in breeding ALB-tolerant cultivars.

## Figures and Tables

**Figure 1 ijms-25-13106-f001:**
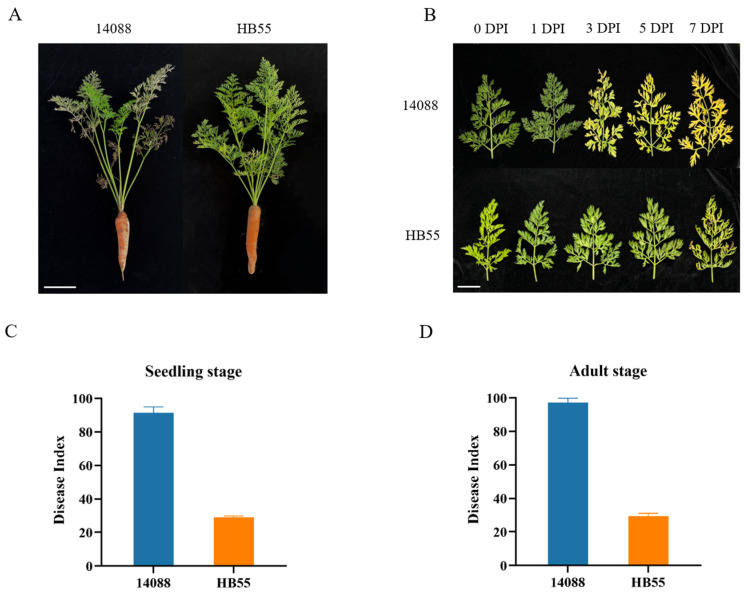
Phenotypic analysis of the response of carrots to *A. dauci* infection. (**A**) ‘14088’ and ‘HB55’ plants in the natural disease nursery. Scale bar = 10 cm. (**B**) Phenotypes of ‘14088’ and ‘HB55’ seedling leaves at 0, 1, 3, 5, and 7 dpi. Scale bar = 5 cm. (**C**) DI of ‘14088’ and ‘HB55’ in the seedling stage. (**D**) DI of ‘14088’ and ‘HB55’ in the adult stage.

**Figure 2 ijms-25-13106-f002:**
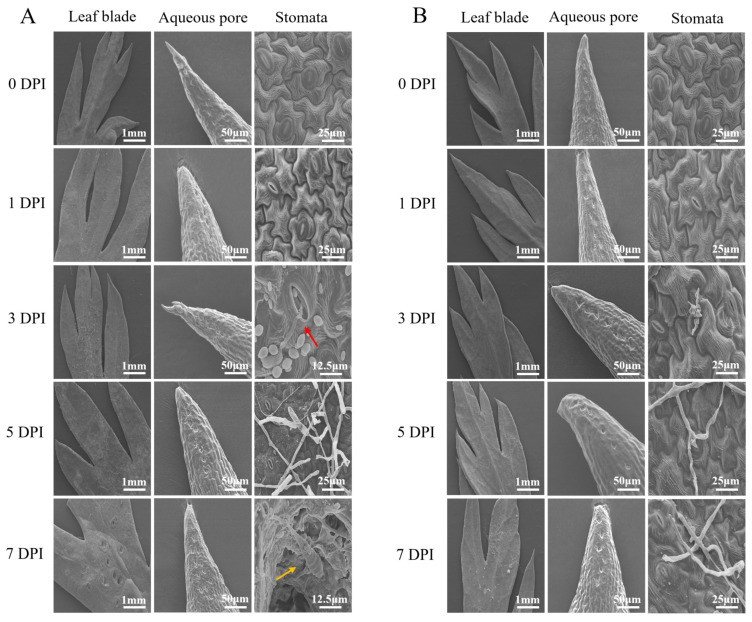
Scanning electron micrographs of leaves of disease-resistant and disease-susceptible cultivars at different stages after inoculation. (**A**) For susceptible cultivar ‘14088’, the red arrow marks the attached spore structure and the yellow arrow marks mature conidia. (**B**) Resistant cultivar ‘HB55’. The leaves are shown at 30×, the aqueous pores are shown at 600×, and the stomata were viewed at 1.2k× and 2.4k×.

**Figure 3 ijms-25-13106-f003:**
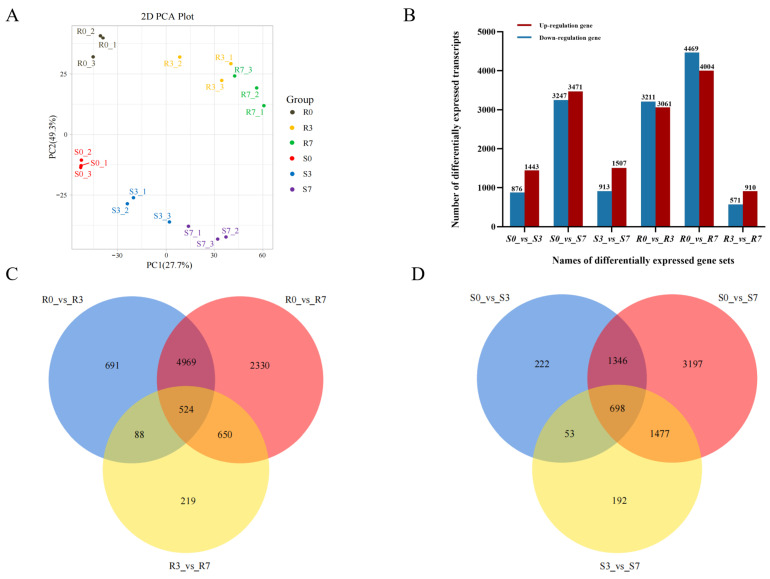
RNA sequencing of carrot cultivars infected with *A. dauci*. (**A**) Principal component analysis (PCA) of the samples. (**B**) The numbers of differentially expressed genes (DEGs) in different comparison groups. (**C**) Venn diagram showing the DEGs present in the resistant cultivar at all three time points. (**D**) Venn diagram showing the DEGs present in the susceptible cultivar at all three time points.

**Figure 4 ijms-25-13106-f004:**
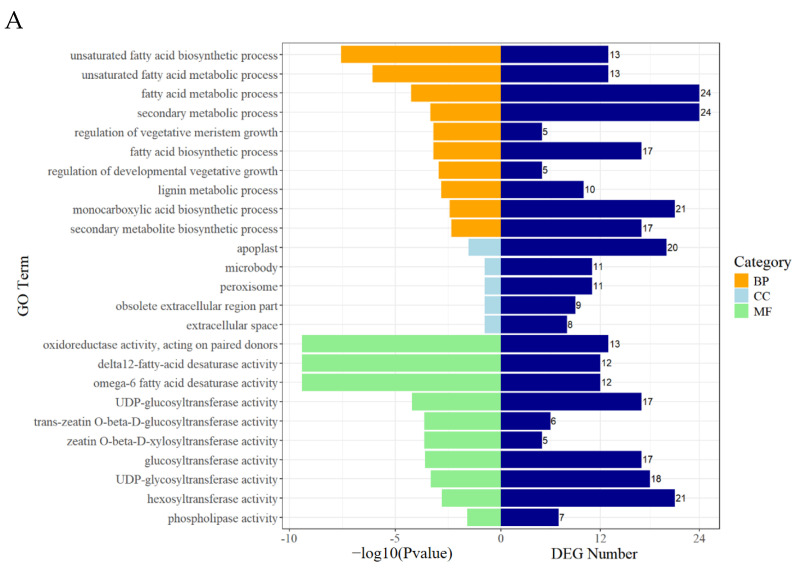
GO and KEGG enrichment analysis of DEGs in ‘HB55’. (**A**) GO enrichment analysis. The left bar represents the −log (*p*-value), and the right bar represents the corresponding number of DEGs. (**B**) KEGG pathway enrichment analysis. Pathways framed in red are closely related to plant disease resistance.

**Figure 5 ijms-25-13106-f005:**
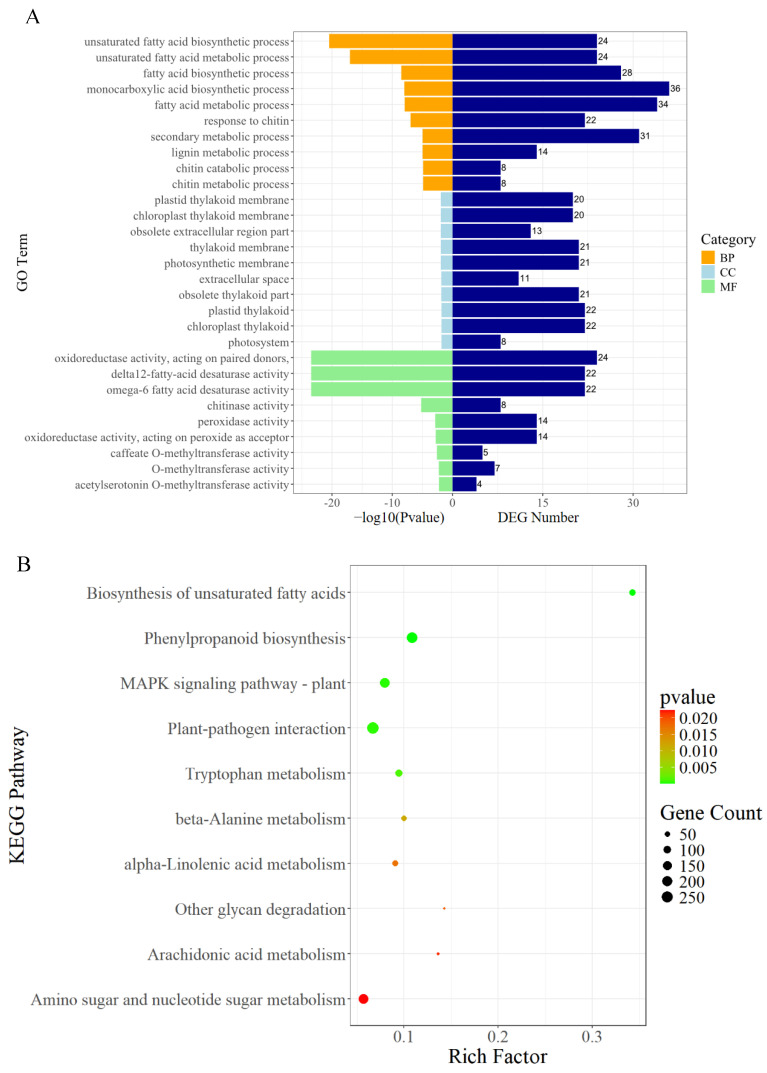
Results of DEG enrichment analysis for the DEGs in ‘14088’. (**A**) Histogram of the results of GO enrichment analysis. (**B**) Scatter plot showing the KEGG pathway enrichment analysis results.

**Figure 6 ijms-25-13106-f006:**
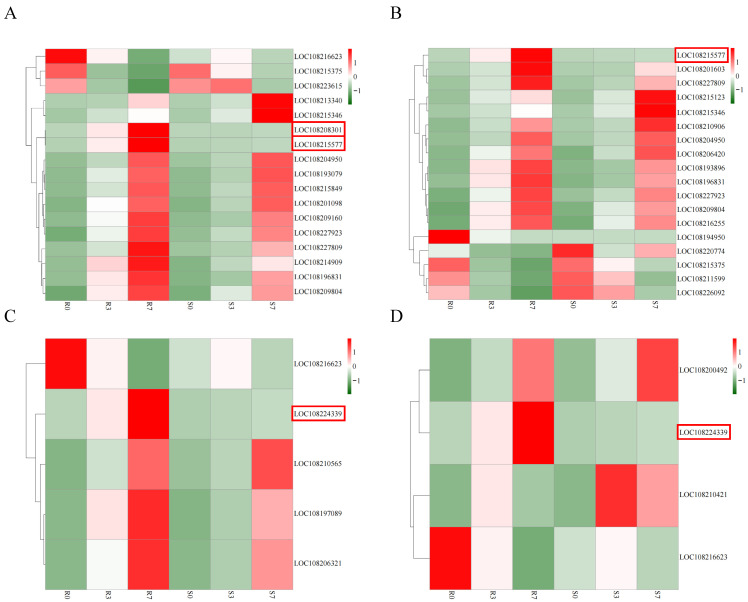
Heatmaps of the genes related to plant disease resistance in ‘HB55’. (**A**) Heatmap of the expression patterns of genes in the secondary metabolic process pathway. (**B**) Heatmap of the expression patterns of genes in the phenylpropanoid biosynthesis pathway. (**C**) Heatmap of the expression patterns of genes in the phenylalanine, tyrosine, and tryptophan biosynthesis pathways. (**D**) Heatmap of the expression patterns of genes in the tyrosine metabolism pathway. The horizontal axis represents the different sample names, and the vertical axis represents the DEGs. Red and green indicate high and low expression, respectively; data of each row was standardized. Genes framed in red are the key resistance response genes.

**Figure 7 ijms-25-13106-f007:**
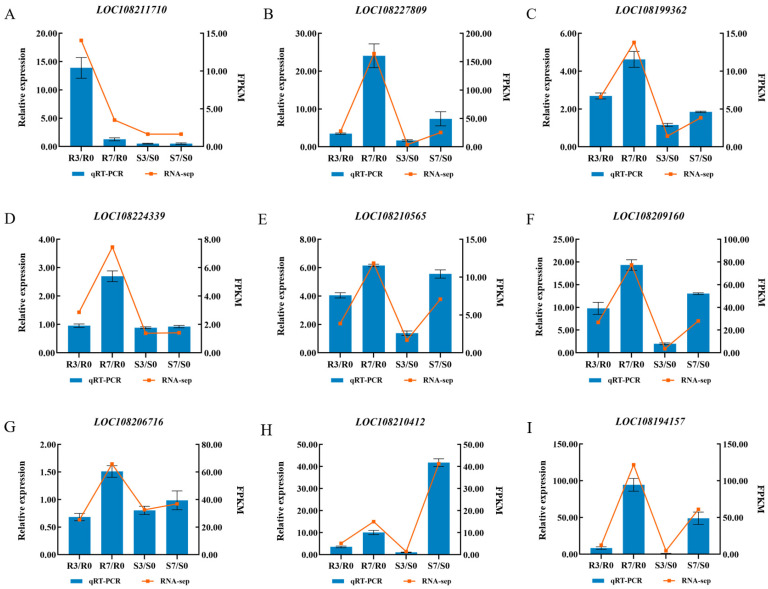
(**A**–**I**) Validation of DEGs by qRT-PCR analysis. The line charts show the gene expression levels determined from the transcriptome analysis (FPKM). The bar graphs represent the qRT-PCR data; the line plots represent the RNA-seq data.

## Data Availability

The Illumina raw sequencing profiles were submitted to the NCBI. BioProject data-base under number PRJNA1194540.

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
