# Peer review of "Transcriptome Analysis Reveals the Molecular Mechanisms of Carrot Adaptation to Alternaria Leaf Blight"

_ijms, 2024, doi:10.3390/ijms252313106_

Round 1
Reviewer 1 Report (New Reviewer)
Comments and Suggestions for Authors
Authors investigated mol mechanism of partial resistance to alternaria in carrot. a susceptible and a partially resistant cv were compared.
The paper is well written and convincing., I have only few comments.
In M+M it is not always clear what samples were taken for the different analyses. authors state that "each experiment consisted of 3 biological replicates"(424). but not clear what this means. first of all I think it was one experiment, in which different measurements were done.
Again in 443 authors mention 3 separate experiments. not sure what that means.. 460: each time point consisted of 3 replicates. not clear what that means. 3 plants? leaves (how many?) from 3 plants? or 3 leaves from 1 plant? 3 plants?
for the reader it is not clear what the content of your samples was. therefore also not clear if they were real biological replicate. Pls try to make this more clear. If you show statistics in figs (eg stand dev or error) you should explain what we see. and also n= ..., and what the number of replicates means. for instance: vert bars show 2x standard error, n=3, each biological replicate consists of a pool of leaves from 3 plants.
57: bactereia?
in fig 7. maybe explain how this was calculated. R3/R) does this maen that the pcr results from day 3 and 0 were used ?
354: With
331 delete THE
350: maybe here explain the "three comparison groups"
Author Response
- In M+M it is not always clear what samples were taken for the different analyses. authors state that "each experiment consisted of 3 biological replicates"(424). but not clear what this means. first of all I think it was one experiment, in which different measurements were done.
Response: Thank you for your suggestion. For disease identification, both artificial inoculation and field nursery testing were conducted in this work. To avoid confusion, we have made supplements to the two methods in M+M in lines 422-433.
- Again in 443 authors mention 3 separate experiments. not sure what that means. 460: each time point consisted of 3 replicates. not clear what that means. 3 plants? leaves (how many?) from 3 plants? or 3 leaves from 1 plant? 3 plants? for the reader it is not clear what the content of your samples was. therefore also not clear if they were real biological replicate. Pls try to make this more clear. If you show statistics in figs (eg stand dev or error) you should explain what we see. and also n= ..., and what the number of replicates means. for instance: vert bars show 2x standard error, n=3, each biological replicate consists of a pool of leaves from 3 plants.
Response: Thank you for your insightful suggestion. We have revised it to “At 0, 1, 3, 5, and 7 dpi, a total of 90 leaf samples were selected from ‘HB55’ and ‘14088’ (9 leaf samples from 3 individual plants for each line at each time point).” in lines 435-436. We also have revised it to “Each time point contained three biological replicates, and each replicate consists of a pool of leaves from 3 plants.” in lines 451-452.
- 57: bactereia?
Response: Thank you for your reminder. We have revised it to “pathogens” (line 56).
- in fig 7. maybe explain how this was calculated. R3/R) does this maen that the pcr results from day 3 and 0 were used?
Response: Thank you for your suggestion. R3/R0 is not a ratio in the figure, it represents the gene expression level at 3dpi relative to 0 dpi in the resistant line. At R3 (3dpi) and R7 (7dpi), the gene expression level from the transcriptome (FPKM) and the qRT-PCR expression levels were calculated as a ratio relative to the expression level of uninfected resistant line R0 (0dpi).
Similar formats can be found in the following literatures: Liu, W.; Wang, X.; Song, L.; Yao, W.; Guo, M.; Cheng, G.; Guo, J.; Bai, S.; Gao, Y.; Li, J. Kang, Z. Comparative Transcriptome and Widely Targeted Metabolome Analysis Reveals the Molecular Mechanism of Powdery Mildew Resistance in Tomato. Int. J. Mol. Sci. 2023, 24, 8236. Yang, C.; Wu, P.; Yao, X.; Sheng, Y.; Zhang, C.; Lin, P.; Wang, K. Integrated Transcriptome and Metabolome Analysis Reveals Key Metabolites Involved in Camellia oleifera Defense against Anthracnose. Int. J. Mol. Sci. 2022, 23, 536.
- 354: With
Response: Thank you for your reminder. We have revised it.
- 331 delete THE
Response: Thank you for your suggestion. We have deleted it.
- 350: maybe here explain the "three comparison groups"
Response: The “three comparison groups” means R0_vs_R3, R0_vs_R7 and R3_vs_R7 comparison groups. We have revised this sentence to “In ‘HB55’, 524 DEGs were identified across the three comparison groups (R0_vs_R3, R0_vs_R7 and R3_vs_R7) and further characterized. Similarly, 698 DEGs were identified in the three ‘14088’ comparison groups (S0_vs_S3, S0_vs_S7 and S3_vs_S7) and further characterized.” (lines 348-351).

Reviewer 2 Report (Previous Reviewer 3)
Comments and Suggestions for Authors
Journal IJMS (ISSN 1422-0067)
Manuscript ID ijms-3319993
Type Article
Title Transcriptome Analysis Reveals the Molecular Mechanisms of Carrot Adaptation to Alternaria Leaf Blight
Authors Chen Liang , Donghang Zhao , Chenggang Ou , Zhiwei Zhao , Feiyun Zhuang * , Xing Liu *
Section Molecular Plant Sciences
Dear Editor
The authors have made the correction in the manuscript correctly.
The text of the manuscript can be proceeded to further editorial stages.
Yours sincerely
Reviewer
I enclose a few remarks, which without my approval should be introduced after the REDACTOR's approval
1.
Figure 2. Please insert bars on each photograph.
2.
Figure 7. Label using letter designations A-F the subsequent diagrams in the figure.
3.
Still the literature list needs to be corrected according to the guidelines for Authors for example item: 19, 33, 36, 53, 58, 63, 65,

Author Response
- Figure 2. Please insert bars on each photograph.
Response: Thank you for your insightful suggestion. We have inserted scale bars on each photograph in Figure 2.
- Figure 7. Label using letter designations A-F the subsequent diagrams in the figure.
Response: Thank you for your suggestion. We have followed your suggestion and used letter designations A-I to indicate individual images in Figure 7.
- Still the literature list needs to be corrected according to the guidelines for Authors for example item: 19, 33, 36, 53, 58, 63, 65,
Response: Thanks for your careful checks, we have revised the content and format of the references.

Reviewer 3 Report (New Reviewer)
Comments and Suggestions for Authors
The work is mainly aimed to explore the molecular mechanisms underlying carrot cultivar adaptation to Alternaria leaf blight, caused by Alternaria dauci.
The technical approach mainly based on transcriptomic analysis is coherent with the aims of the research.
Results provide interesting insights about the genes involved in the resistance of carrot against A.dauci
The following revisions could improve the quality of the work:
• In the “Introduction” section,
o In order to make the content more tailored on carrot and its resistance mechanisms against A.dauci, some interesting and more specific studies could be added, such as https://link.springer.com/article/10.1007/s10658-007-9241-6 and https://www.nature.com/articles/s41598-018-31700-2
• In the “Results” section,
§ Correlation heatmaps are illustrated and discussed, but in “Material and Methods” section the method used to develop them is not clear. Please explain better.
• The ‘Discussion’ section is dispersive. I suggest to focus more on the achieved results.
• In the “Conclusions” section,
o the value added of the research compared to previous studies could be underlined.
Some minor issues:
• A table of abbreviations used in the manuscript is suggested. It can be included at the end of the manuscript.
• Please check the references format, https://www.mdpi.com/journal/ijms/instructions
Author Response
- In the “Introduction” section: In order to make the content more tailored on carrot and its resistance mechanisms against A.dauci, some interesting and more specific studies could be added, such as https://link.springer.com/article/10.1007/s10658-007-9241-6 and https://www.nature.com/articles/s41598-018-31700-2
Response: This is a very insightful suggestion. In lines 35-39 and 65-74. we have added more references to more specific studies on resistance mechanisms against A. dauci in the INTRODUCTION part of the revised manuscript.
- In the “Results” section: Correlation heatmaps are illustrated and discussed, but in “Material and Methods” section the method used to develop them is not clear. Please explain better.
Response: We think this is an excellent suggestion. We have added an additional description of the method used to develop correlation heatmaps in lines 472-480.
- The ‘Discussion’ section is dispersive. I suggest to focus more on the achieved results.
Response: We sincerely appreciate the valuable comment. The ‘Discussion’ section of the revised manuscript has been improved as suggested.
- In the “Conclusions” section, the value added of the research compared to previous studies could be underlined.
Response: Thank you for your insightful suggestion. We have added an additional description in lines 501-503.
- A table of abbreviations used in the manuscript is suggested. It can be included at the end of the manuscript.
Response: Thank you for your insightful suggestion. We have added a table of abbreviations used in the revised manuscript in line 514.
- Please check the references format, https://www.mdpi.com/journal/ijms/instructions
Response: Thanks for your careful checks, we are sorry for our carelessness. We have checked and revised the content and format of all references.

This manuscript is a resubmission of an earlier submission. The following is a list of the peer review reports and author responses from that submission.
Round 1
Reviewer 1 Report
Comments and Suggestions for Authors
The proposed manuscript shows a contribution to the understanding of carrot resistance to Alternaria disease.
The title is clear and meaningful. It adequately reflects the subject matter.
The abstract summarises the study and the main results well. However, it lacks a description of the practical importance of the finding for breeding programmes.
The introduction clearly states the relevance of carrot as a crop, the problem of Alternaria leaf blight (ALB) and the potential of transcriptomic technologies to address this problem. However, detailed references to recent technological developments in plant resistance analysis to fungal diseases are lacking.
The authors should include a broader discussion of the increasing role of transcriptomic and genomic techniques in breeding programmes, with citations of recent work beyond 2020. It would be useful to include a reference to the importance of improving ALB resistance to reduce fungicide use.
The methodology is well explained, with clear descriptions of inoculation experiments, RNA-seq analysis and the use of SEM (scanning electron microscopy).
However, I must point out that details of some critical steps are missing, such as the rationale for selecting samples for transcriptomic analysis. There is also no discussion of how the inoculation methods or SEM image analysis were standardised, which could affect the reproducibility of the results.
The phenotyping results (lines 86-112) clearly show differences between resistant and susceptible cultivars, with data supported by well presented phenotypic photographs.
However, it would be better to include tables with quantitative data to describe disease progression more precisely, e.g. by measuring lesion size or mycelium growth rate.
Regarding SEM observations (lines 113-126), SEM images provide useful visual information on fungal colonisation, but it would be appropriate to provide a quantitative discussion based on SEM images, e.g. an analysis of mycelium density or a comparative assessment of observed damage.
The transcriptomic analysis (lines 140-266) is well presented, with key findings on **differentially expressed genes (DEGs)** and enrichment of relevant pathways (phenylpropanoids, tyrosine metabolism). The immediate practical implications of these DEGs for breeding or agriculture are not discussed.
The transcriptomic data are well described, but the part on how these genes could be validated in field trials is missing.
The authors should provide a more detailed discussion of the potential use of these genes in resistant breeding programmes and suggest possible future studies that could confirm the functionality of the genes through functional testing or gene editing.
The discussion (lines 267-337) adequately interprets the results obtained and relates them to the resistance of other plants and biosynthetic pathways, but does not sufficiently address the real agricultural context, such as the implications for cultivation in different environments or under different climatic stresses.
The authors should add a section discussing the relevance of these results for real agronomic practice, including field trials under different climatic conditions or in areas with high ALB prevalence.
The conclusion summarises the main results and the importance of the research well, but lacks a clear projection of how these results can be used to improve resistant carrot varieties.
It would be better to add a final statement highlighting the potential use of these results in future breeding programmes and the possibility of developing new resistant varieties through biotechnological approaches.
Overall, the manuscript is a valuable and thorough contribution to the understanding of resistance to Alternaria disease in carrots. With improvements in the practical discussion, conclusions and quantitative analysis of phenotypic and transcriptomic data, the manuscript could have an even greater impact on both agricultural research and practical applications.
Author Response
- The abstract summarises the study and the main results well. However, it lacks a description of the practical importance of the finding for breeding programmes.
Response: Thank you for your suggestion, we have added this aspect in the abstract: “These findings provide insights into the resistance mechanism of carrots to ALB, and provide key candidate genes and expression regulation information for molecular breeding of carrot disease resistance” in lines 21-23.
- The introduction clearly states the relevance of carrot as a crop, the problem of Alternaria leaf blight (ALB) and the potential of transcriptomic technologies to address this problem. However, detailed references to recent technological developments in plant resistance analysis to fungal diseases are lacking.
Response: The application of related technologies in plant resistance analysis has been supplemented in lines 63-72.
- The authors should include a broader discussion of the increasing role of transcriptomic and genomic techniques in breeding programmes, with citations of recent work beyond 2020. It would be useful to include a reference to the importance of improving ALB resistance to reduce fungicide use.
Response: To enhance the important role of omics analysis, 6 relevant literatures beyond 2020 were added. Also in lines 303-304, regarding “the importance of improving ALB resistance to reduce fungicide use”, add the following: “And it cannot be ignored, the cultivation of disease resistant varieties will also reduce the damage of environment, caused by the application of a large number of chemical fungicides”.
- The methodology is well explained, with clear descriptions of inoculation experiments, RNA-seq analysis and the use of SEM (scanning electron microscopy).
However, I must point out that details of some critical steps are missing, such as the rationale for selecting samples for transcriptomic analysis. There is also no discussion of how the inoculation methods or SEM image analysis were standardised, which could affect the reproducibility of the results.
Response: More detailed experimental procedure for sample selection and microscopic observation by SEM have been supplemented in lines 431-432, 438-453.
- The phenotyping results (lines 86-112) clearly show differences between resistant and susceptible cultivars, with data supported by well presented phenotypic photographs. However, it would be better to include tables with quantitative data to describe disease progression more precisely, e.g. by measuring lesion size or mycelium growth rate.
Response: Thank you for your suggestion. We have added experimental data about this part and described them in lines 114-116.
- Regarding SEM observations (lines 113-126), SEM images provide useful visual information on fungal colonisation, but it would be appropriate to provide a quantitative discussion based on SEM images, e.g. an analysis of mycelium density or a comparative assessment of observed damage.
Response: Thank you for your suggestion. We have added experimental data about this part and described them in lines 151-152.
- The transcriptomic analysis (lines 140-266) is well presented, with key findings on **differentially expressed genes (DEGs)** and enrichment of relevant pathways (phenylpropanoids, tyrosine metabolism). The immediate practical implications of these DEGs for breeding or agriculture are not discussed.
Response: In the discussion section (lines 393-395), we have added the role of candidate DEG genes in future breeding applications.
- The transcriptomic data are well described, but the part on how these genes could be validated in field trials is missing.
Response: This work aims to explore the relevant pathways and genes involved in the carrot disease resistance network. In the future, to achieve breeding applications, these candidate genes will be validated in more carrot materials to clarify their functions, and facilitate their transfer and utilization.
- The authors should provide a more detailed discussion of the potential use of these genes in resistant breeding programmes and suggest possible future studies that could confirm the functionality of the genes through functional testing or gene editing.
Response: Thank you for your suggestion, the potential use of these genes in future studies were added in lines 393-395.
- The discussion (lines 267-337) adequately interprets the results obtained and relates them to the resistance of other plants and biosynthetic pathways, but does not sufficiently address the real agricultural context, such as the implications for cultivation in different environments or under different climatic stresses.
Response: Yes, we have supplemented these contents in the discussion (lines 298-302).
- The authors should add a section discussing the relevance of these results for real agronomic practice, including field trials under different climatic conditions or in areas with high ALB prevalence.
Response: We have supplemented these contents in the discussion (lines 297-305).
- The conclusion summarises the main results and the importance of the research well, but lacks a clear projection of how these results can be used to improve resistant carrot varieties. It would be better to add a final statement highlighting the potential use of these results in future breeding programmes and the possibility of developing new resistant varieties through biotechnological approaches.
Response: The role of these results in subsequent variety breeding has been emphasized in lines 488-497.

Reviewer 2 Report
Comments and Suggestions for Authors
The current paper devoted to investigation of the molecular mechanism of carrot resistance to leaf blight infection. Authors used modern molecular methods and proposed possible mechnaism of resistance to this infection. Authors claimed that secondary metabolites, phenylpropanoid pathway and tyrosine metabolism may play a key role in planta adaptation process.
Line 56: in majority of the case transcriptome provide only adaptation mechanism, without epigenetic and physiological mechanism “he key molecular mechanisms of disease resistance”.
Line 82: “investigate the resistance response of carrots to ALB by RNA sequencing” ¿? Please, re-formulate better. Resistant response by RNA sequencing?
Lines 169 – 170_ 3dpi maybe too later stage. And DEG may reflected rather secondary response/adaptation, but not primary one. At 3 dpi line 14088 already have damage cells. The most interesting will be first few hours.
You need to mention in discussion that DEG reflected not response, but adaptation.
Moreover, I am sure that infection change local hormonal signaling (auxin, for example).
This can not be detected by total RNA.
Please, add this possible mechanism to discussion.
Moreover, it will be nice to discuss also possible differences in cell wall properties between lines.
Line 305: “ differential transcriptomic responses” ¿? Transcriptomic can not have a response, transcriptomic can have an abundance.
Line 358: light intensity?
Fig 1. scale bar?
Comments on the Quality of English Languagesome sentences not in scientfic style, edition are required
Author Response
- Line 56: in majority of the case transcriptome provide only adaptation mechanism, without epigenetic and physiological mechanism “the key molecular mechanisms of disease resistance”.
Response: Yes, we have revised it.
- Line 82: “investigate the resistance response of carrots to ALB by RNA sequencing”? Please, re-formulate better. Resistant response by RNA sequencing?
Response: Thank you for your suggestion. We have revised it.
- Lines 169-170 3 dpi maybe too later stage. And DEG may reflected rather secondary response/adaptation, but not primary one. At 3 dpi line 14088 already have damage cells. The most interesting will be first few hours.
Response: Yes, early expression changes are important. This study focuses on the dynamic gene expression during the development of the disease, identifying the genes and pathways that play a role throughout the entire disease progression process.
- You need to mention in discussion that DEG reflected not response, but adaptation. Moreover, I am sure that infection change local hormonal signaling (auxin, for example). This can not be detected by total RNA. Please, add this possible mechanism to discussion.
Response: Yes, more comprehensive discussions was conducted around the hormonal signaling and the other pathways in lines 360-370.
- Moreover, it will be nice to discuss also possible differences in cell wall properties between lines.
Response: Thank you for your suggestion. We have added relevant discussions for this in lines 370-379.
- Line 305: “differential transcriptomic responses”? Transcriptomic can not have a response, transcriptomic can have an abundance.
Response: Thank you for your suggestion. We have revised it.
- Line 358: light intensity?
Response: Light intensity is 200~400 μmol m-2 s-1 photosynthetic photon flux density (PPFD). We have added this section to lines 416-417.
- Fig 1. scale bar?
Response: Thank you for your suggestion, we have modified the scale bar in Figure 1 to make it more visible. And the size of the scale bar has been added to the figure notes.
- some sentences not in scientfic style, edition are required.
Response: Thank you for your suggestion, the revised manuscript has been polished by a professional language editor.

Reviewer 3 Report
Comments and Suggestions for Authors
Journal IJMS (ISSN 1422-0067)
Manuscript ID ijms-3248748
Type Article
Title Transcriptome Analysis Reveals the Resistance Response of Carrot (Daucus carota L.)
to Alternaria Leaf Blight
Authors Chen Liang , Donghang Zhao , Chenggang Ou , Zhiwei Zhao , Feiyun Zhuang * ,
Xing Liu *
Section Molecular Plant Sciences
Dear Editor
The text of the manuscript requires additions.
Please find below my comments
With best regards
Reviewer
TITLE
1.
Please rethink the subject line.
2.
Please provide the correct species name in English instead of the generic name.
ABSTRACT
3.
Please check step by step in the abstract that all structural elements of the abstract have been
followed:
(i) put the introduction discussed in a broad context
(ii) highlight the purpose of the study
(iii) briefly describe the main methods used
(iv) summarise the main relevant results
(v) indicate the main conclusions.
KEWORDS
4.
Please eliminate the following terms in the title of the manuscript.
INTRODUCTION
l. 23 - 29.
5.
Please provide the species name in English instead of the genus
6.
Please explain, not every audience is into this subject matter
7.
Please provide the name of the family in Latin.
8.
Please provide figures and justification:
a. ...widely grown worldwide...
b. ...as one of the top ten vegetable crops ... .
9.
Please provide figures, values, concentrations etc. When citing four items of literature, the
reader expects factual knowledge in return for mental abbreviations.
10.
l. 27-29. Please provide up-to-date figures e.g. area under cultivation , estimated yield etc. .
l. 30-37 .
11.
Please state what in figures the problem caused by Alternaria e.g. losses caused , attempts to
protect etc .
l. 38-50.
12.
Please state the disrupted metabolic processes in plants caused by Alternaria.
13.
Please complete the anatomical and ultrastructural changes caused by Alternaria.
l. 51-74.
14.
l.54-56. Please explain, elaborate on the information with reference to the manuscript topic
...immune response, signal transduction, and defence metabolism... .
15.
l. 64-65. please provide these metabolic pathways and transformations.
16.
l. 68-70. Please state these genes and whether they are compressed with reference to the
manuscript theme.
17.
Please give examples from related plants, posted information on tomato and strawberry (these
are genera from a distant systematic position relative to Daucus carota).
18.
When first used, the abbreviation should be expanded
...P. cactorum... .
l. 75-84.
19.
Please reinforce the rationale for the research undertaken.
20.
Please specify the precise, specific aim of the work corresponding to the next methodological
steps.
l. 82-84.
21.
Sentence not in this place.
RESULTS
22.
Please eliminate multiple quotations of the same figure.
23.
Such not in the text, should be in the marks in the figure description.
... (yellow arrow in Figure 2A)....
...red arrow in Figure 2A)....
FIGURE 3B.
24.
Insert X-axis designation
FIGURE 7.
25.
Insert y-axis and X-axis label.
DISCUSSION
26.
l. 270- 297. First the trait studied, a parameter in your own research, and then a discussion
with the results of other authors.
27.
Please reword the discussion text. Change the nature of the review to a discussion.
28.
Please state the applicability of the research results obtained.
29.
Please complete the perspective of the research in the future .
MATERIALS AND METHODS
30.
Please provide literature for each methodological step to support the validity of the correct
choice.
31.
Please complete the methodological information in each sub-step so that the experiment can
be carried out again.
32.
l. 373-380 The scanning microscopy methodology is more complicated than the authors give.
The methodology requires additions
33.
Please complete:
(i) concentrations for each solution,
(ii) time,
(iii) temperature,
(iv) in the dehydration process, please give successive concentrations
(v) in which apparatus the drying was done, conditions, etc.
(vi) in what apparatus the gold was sputtered
34.
Please state the scheme adopted during scanning microscope observation to eliminate
accidental and erroneous shots.
35.
Please indicate the number of slides prepared (from how many plants, number of fragments,
place of collection, number of observations.
CONCLUSION
36.
Please formulate a specific conclusion from each methodological step instead of e.g..
... In this work, we identified a germplasm resource for ALB resistance and clarified the
invasion and colonization modes of A. dauci...
REFERENCE
37.
Please make a correction according to the guidelines for Authors e.g.
38.
Item 2; 3; 4; 5; 6 etc. it is impossible to list them all

Author Response
TITLE
- Please rethink the subject line. “…the Resistance Response…”
Response: Thank you for your suggestion, we have revised the title to “Transcriptome Analysis Reveals the Molecular Mechanisms of Carrot Adaptation to Alternaria Leaf Blight” in lines 2-3.
- Please provide the correct species name in English instead of the generic name.
Response: Yes, we have revised it in the title.
ABSTRACT
- Please check step by step in the abstract that all structural elements of the abstract have been followed:
(i) put the introduction discussed in a broad context
(ii) highlight the purpose of the study
(iii) briefly describe the main methods used
(iv) summarise the main relevant results
(v) indicate the main conclusions.
Response: We have refined the abstract and checked it in detail to ensure that it has complete structural elements. The re-edited abstract as follows: Carrot (Daucus carota L.) is an important vegetable crop that is rich in carotenoids and is widely cultivated throughout the world. Alternaria leaf blight (ALB), caused by infection with Alternaria dauci (A. dauci), is the most serious fungal disease in carrot production. Although several quantitative trait loci associated with ALB resistance have been identified, the genetic mechanisms underlying this resistance remain largely unelucidated. The aim of the present study was to clarify the infection mode of A. dauci and examine the molecular mechanisms underlying carrot cultivar adaptation to ALB by RNA sequencing. In this study, microscopic observation revealed A. dauci invades leaf tissues by entering through stomata, and resistant germplasms may significantly inhibit the infection and colonization of A. dauci. In addition, transcriptomic analyses were performed to detect the key pathways and genes associated with the differential responses between ALB-resistant (HB55) and ALB-susceptible (14088) carrot cultivars. These results suggest that secondary metabolic biosynthesis, phenylpropanoid biosynthesis, and tyrosine metabolism might play important roles in the prevention of carrots to A. dauci. Three candidate genes (LOC108208301, LOC108215577, and LOC108224339) that were specifically upregulated in the resistant carrot cultivar ‘HB55’ after A. dauci infection were identified as the key resistance response genes. These findings provide insights into the resistance mechanism of carrots to ALB, and provide key candidate genes and expression regulation information for molecular breeding of carrot disease resistance.
KEWORDS
- Please eliminate the following terms in the title of the manuscript.
Response: The following terms in the title were eliminated and new keywords were provided as follow: Daucus carota; adaptation mechanism; resistance gene; transcriptome.
INTRODUCTION
- Line 23 - 29. Please provide the species name in English instead of the genus. “…Carrot…”
Response: Thank you for your suggestion, the species name in English is carrot as stated in this manuscript and other articles: Transcriptome Sequencing Reveals the Mechanism of Auxin Regulation during Root Expansion in Carrot (Li et al., 2024), Response of Carrot (Daucus carota L.) to Multi-Contaminated Soil from Historic Mining and Smelting Activities (Novák et al., 2023).
- Please explain, not every audience is into this subject matter. “…2n=2x=18…”
Response: The “2n” means that the carrot is a diploid plant, “X” is the number of chromosomes in each chromosome set, and finally “X=9” is a chromosome set containing 9 chromosomes. We have removed this part of the manuscript to make it easier for more people to understand.
- Please provide the name of the family in Latin.
Response: Adoption of this name change will take time and many readers will continue to recognize Umbelliferae rather than Apiaceae (previously known as Umbelliferae), and we have revised it in line 27.
- Please provide figures and justification: a ...widely grown worldwide...b ...as one of the top ten vegetable crops ...
Response: To prevent misunderstandings, we have revised this sentence to “and widely grown in Asia, Europe and the Americas” (line 28).
- Please provide figures, values, concentrations etc. When citing four items of literature, the reader expects factual knowledge in return for mental abbreviations.
Response: Thank you for your suggestion, we have removed this section “In addition, natural polyacetylenol, a product extracted from carrots, has the potential to delay aging and increase energy metabolism” to avoid misunderstanding. And re-write the first paragraph of the introduction section.
- Line 27-29. Please provide up-to-date figures e.g. area under cultivation, estimated yield etc .
Response: The latest data of planting areas has been supplemented in lines 29-33.
- Line 30-37. Please state what in figures the problem caused by Alternaria e.g. losses caused, attempts to protect etc.
Response: In severe cases, ALB resulting in a 40-60% yield reduction in carrot, and all production areas in the northern provinces of China were affected by ALB. A great deal of chemical pesticides are applied to control this disease, causing the problem of environmental pollution and endangering the edible safety of the consumers. The situation has been supplemented in lines 37-43.
- Line 38-50. Please state the disrupted metabolic processes in plants caused by Alternaria.
Response: Thank you for your suggestion. The disrupted metabolic processes in plants caused by Alternaria has been supplemented in lines 49-50.
- Line 51-74. Please complete the anatomical and ultrastructural changes caused by Alternaria.
Response: Thank you for your suggestion, we have added an additional description in lines 55-58.
- Line 54-56. Please explain, elaborate on the information with reference to the manuscript topic “...immune response, signal transduction, and defence metabolism...”.
Response: This sentence may have been misunderstood, we have revised it in lines 74-75.
- Line 64-65. Please provide these metabolic pathways and transformations.
Response: Thank you for your suggestion, we have added an additional description in lines 83-85.
- Line 68-70. Please state these genes and whether they are compressed with reference to the manuscript theme.
Response: Thank you for your suggestion, we have added an additional description in lines 90-91.
- Please give examples from related plants, posted information on tomato and strawberry (these are genera from a distant systematic position relative to Daucus carota).
Response: Yes, it’s better to provide the examples from related plants, however, there is relatively little transcriptomic research on Umbelliferae crops, and even fewer reports on disease resistance. Therefore, the related studies in tomato and strawberry were selected.
- When first used, the abbreviation should be expanded “...P. cactorum...”.
Response: Thank you for your suggestion. We have revised it.
- Line 75-84. Please reinforce the rationale for the research undertaken.
Response: We have added the rationale in lines 98-101.
- Please specify the precise, specific aim of the work corresponding to the next methodological steps.
Response: We have added the rationale in lines 98-106.
- Line 75-84. Sentence not in this place.
Response: We have deleted this sentence.
RESULTS
- Please eliminate multiple quotations of the same figure.
Response: Thank you for your suggestion. We have deleted it.
- Such not in the text, should be in the marks in the figure description “.... (yellow arrow in Figure 2A)....” “...red arrow in Figure 2A)...”.
Response: We have transferred them to the figure description.
- FIGURE 3 B. Insert X-axis designation.
Response: We have added X-axis designation for FIGURE 3 B.
- FIGURE 7. Insert y-axis and X-axis label.
Response: We have added these parts to FIGURE 7. Reference was made to the figures in the following literatures: Liu, W.; Wang, X.; Song, L.; Yao, W.; Guo, M.; Cheng, G.; Guo, J.; Bai, S.; Gao, Y.; Li, J. Kang, Z. Comparative Transcriptome and Widely Targeted Metabolome Analysis Reveals the Molecular Mechanism of Powdery Mildew Resistance in Tomato. Int J Mol Sci. 2023, 24, 8236. Yang, C.; Wu, P.; Yao, X.; Sheng, Y.; Zhang, C.; Lin, P.; Wang, K. Integrated Transcriptome and Metabolome Analysis Reveals Key Metabolites Involved in Camellia oleifera Defense against Anthracnose. Int J Mol Sci. 2022, 23, 536.
DISCUSSION
- Line 270-297. First the trait studied, a parameter in your own research, and then a discussion with the results of other authors.
Response: In lines 297-315, we have revised these sentences to better align with the logic of a discussion.
- Please reword the discussion text. Change the nature of the review to a discussion.
Response: We have reorganized the discussion section of the revised manuscript.
- Please state the applicability of the research results obtained.
Response: We have supplemented the applicability of the research results in discussion section (lines 393-396).
- Please complete the perspective of the research in the future.
Response: In lines 393-396, we have completed “In the future, targeting these candidate genes and key regulatory genes in related pathways, conducting deserves gene function analysis and molecular marker development will accelerate the process of molecular improvement of carrot variety resistance to ALB”.
MATERIALS AND METHODS
- Please provide literature for each methodological step to support the validity of the correct choice.
Response: The necessary literature has been supplemented in line 401, 413, and 429.
- Please complete the methodological information in each sub-step so that the experiment can be carried out again.
Response: Yes, we have supplemented more detailed description of the experiments.
- Line 373-380. The scanning microscopy methodology is more complicated than the authors give. The methodology requires additions.
Response: In lines 438-453, more detailed experimental procedure for scanning microscopy have been supplemented.
33.
Please complete:
(i) concentrations for each solution,
(ii) time,
(iii) temperature,
(iv) in the dehydration process, please give successive concentrations
(v) in which apparatus the drying was done, conditions, etc.
(vi) in what apparatus the gold was sputtered
Response: Thank you for your detailed suggestion, we have supplemented each of these experimental conditions one by one.
- Please state the scheme adopted during scanning microscope observation to eliminate accidental and erroneous shots.
Response: Thank you for your suggestion. The observation scheme was supplemented in lines 448-453.
- Please indicate the number of slides prepared (from how many plants, number of fragments, place of collection, number of observations.
Response: We have supplemented these numbers in lines 448-453.
CONCLUSION
- Please formulate a specific conclusion from each methodological step instead of e.g “... In this work, we identified a germplasm resource for ALB resistance and clarified the invasion and colonization modes of A. dauci...”
Response: Thank you for your suggestion, we have have reorganized the conclusion section in lines 488-497.
REFERENCE
- Please make a correction according to the guidelines for Authors e.g.
Response: We have revised the content and format of the references.
- Item 2; 3; 4; 5; 6 etc. it is impossible to list them all
Response: Thank you for your suggestion, we have revised the content and format of the references.

Round 2
Reviewer 2 Report
Comments and Suggestions for Authors
Thank you! The text is OK, correction are completed. Some polishing may require during final adjustement.
It will be great to increase contrast on figure 3.
Author Response
1. It will be great to increase contrast on figure 3.
Response: This is a very insightful suggestion. We have modified Figure 3 in the revised manuscript to make it easier to distinguish between the groups by colour. In addition, we truly appreciate your high level academic insights once again.